# Role of Sirtuins in the Pathobiology of Onco-Hematological Diseases: A PROSPERO-Registered Study and In Silico Analysis

**DOI:** 10.3390/cancers14194611

**Published:** 2022-09-23

**Authors:** João Vitor Caetano Goes, Luiz Gustavo Carvalho, Roberta Taiane Germano de Oliveira, Mayara Magna de Lima Melo, Lázaro Antônio Campanha Novaes, Daniel Antunes Moreno, Paola Gyuliane Gonçalves, Carlos Victor Montefusco-Pereira, Ronald Feitosa Pinheiro, Howard Lopes Ribeiro Junior

**Affiliations:** 1Center for Research and Drug Development (NPDM), Federal University of Ceara, Fortaleza 60020-181, Ceará, Brazil; 2Post-Graduate Program of Pathology, Federal University of Ceara, Fortaleza 60020-181, Ceará, Brazil; 3Post-Graduate Program in Translational Medicine, Federal University of Ceara, Fortaleza 60020-181, Ceará, Brazil; 4Post-Graduate Program in Medical Science, Federal University of Ceara, Fortaleza 60020-181, Ceará, Brazil; 5Molecular Oncology Research Center, Barretos Cancer Hospital, Barretos 14784-400, São Paulo, Brazil; 6Boehringer Ingelheim Pharma GmbH & Co. KG, Pharmaceutical Development Biologicals, 88400 Biberach an der Riss, Germany

**Keywords:** sirtuins, onco-hematological disorders, aging biomarkers

## Abstract

**Simple Summary:**

The aging of the hematological system can cause physiological disorders such as anemia, reduced immunity, and the increased incidence of blood cancer. Patients diagnosed with hematologic malignancies comprise nearly 10% of all cancer deaths identified in international epidemiologic studies. Therefore, it is considered a public health problem worldwide. Scientific evidence demonstrates the important involvement of sirtuins (SIRTs) in the pathogenesis of several types of solid tumors. However, the role of SIRTs in the pathobiology of malignant hematological diseases has not yet been systematically reviewed. In this systematic review, we highlight the role of different SIRTs in the pathogenesis of acute and chronic leukemias, lymphoma and myeloma. Also, we performed a bioinformatic analysis to identify whether the expression of SIRTs is altered in onco-hematological diseases, such as lymphomas and leukemias. The advent of new applicability of SIRTs in the process of aging and hematological carcinogenesis may allow the development of new diagnostic and therapeutic approaches for these diseases.

**Abstract:**

The sirtuins (*SIRT*) gene family (*SIRT1* to *SIRT7*) contains the targets implicated in cellular and organismal aging. The role of SIRTs expression in the pathogenesis and overall survival of patients diagnosed with solid tumors has been widely discussed. However, studies that seek to explain the role of these pathways in the hematopoietic aging process and the consequences of their instability in the pathogenesis of different onco-hematological diseases are still scarce. Therefore, we performed a systematic review (registered in PROSPERO database #CRD42022310079) and in silico analysis (based on GEPIA database) to discuss the role of SIRTs in the advancement of pathogenesis and/or prognosis for different hematological cancer types. In summary, given recent available scientific evidence and in silico gene expression analysis that supports the role of SIRTs in pathobiology of hematological malignances, such as leukemias, lymphomas and myeloma, it is clear the need for further high-quality research and clinical trials that expands the SIRT inhibition knowledge and its effect on controlling clonal progression caused by genomic instability characteristics of these diseases. Finally, SIRTs represent potential molecular targets in the control of the effects caused by aging on the failures of the hematopoietic system that can lead to the involvement of hematological neoplasms.

## 1. Introduction

Cancer is a multifactorial disease related to the process of aging, as the incidence of most cancers increases with age following an accumulation of mutations and genomic instability. Aging of the hematological system can cause physiological disorders such as anemia, reduced immunity, and the increased incidence of blood cancer. Hematopoietic stem cells (HSCs) play an important role in the process of hematologic carcinogenesis and its pluripotency efficiency decreases significantly with aging [1,2].

Stem cells have de novo abilities during differentiation that includes hematopoietic stem cells (HSCs) towards myeloid or lymphoid-lineage cells. During senescence, HSCs may present reduced functionality towards hematological maturation, and this leads to deficiencies in all blood cells. This has been considered mostly from the recurrent mutations in genes that encode the regulators of the chromatin structure [3,4].

Nonetheless, mutations on driver genes may compromise HSCs lineage and this is linked to aging as well. When a substantial number of HSCs derives to one dominant lineage (particularly the myeloid), this is called clonal hematopoiesis (CH) [5]. One of the main phenotypical characteristics of CH is risk for hematological malignancy [6,7]. Through the aging process, the dominance of the myeloid lineage in CH may increase mutations in somatic cells generating genomic instability as increased DNA damage or deficiency in DNA repair. This is associated with the replicative stress linked to aging [8].

The relationship and influence of aging in cellular processes includes studies on homeostasis, tissue regeneration and cancer control towards suppression, regression, and maintenance of cell viability [9,10]. Corradi et al. [11] showed that mesenchymal stem cells (MSCs) from hematological disorders support the viability of leukemic cells, even if these MSCs are not phenotypically different from the healthy MSCs. This was observed in MSCs from patients with myelodysplastic syndrome (MDS) and acute myeloid leukemia (AML).

The HSCs are in bone marrow microenvironment/niche (BMN) in which they are connected to other cells or molecular components. In human aging, factors such as increased angiogenesis, particularly because of co-cultures of the BMN, are responsible for altering the stroma cell compartment. In skewed MSCs, the accumulation of adipocytes and loss of nerve fibers might induce niche aging, and epigenetic regulators, such as the Sirtuin protein, may prevent this aging [7,12,13]. In mice, point mutations of the beta-catenin protein in osteoblasts altered the differentiation potential of MSCs via NOTCH-1 signaling. This feedback mechanism is prevalent in 38% of AML patients and stimulates leukemogenesis [14].

In mammals, the Sirtuins (SIRTs) have extremely conserved expression from *SIRT1* to *SIRT7*. The overexpression of SIRTs is associated with increased cellular survival and are reported to be linked to the deacetylation of lysine moieties [15]. SIRTs are homologous to the gene *Sir2* (*Silent information regulator-2*) and are NAD+ hydrolysis-dependent. SIRTs are particularly important in disease prevention and in reducing the effects of aging and cancer [15,16,17]. Attached to the effects of aging, the production of NAD+ tends to be decreased, which reflects on the poor development of dependent processes. However, studies identifying the precursors of NAD+ can restore the effect of SIRTs in aged organisms [18,19,20]. Figure 1 describes the biochemical process that guides the synthesis of NAD+ and its deacetylation activity together with the SIRTs.

SIRTs are classified as Histone Deacetylases (HDACs) of class 3 and act on several cellular compartments. For example, SIRT1, SIRT6, and SIRT7 present activity in the cell nucleus, while SIRT2 can be found in the cytoplasm. SIRT3, SIRT4 and SIRT5 are associated with main activity in the mitochondria [16,21]. SIRTs are found in several regulation pathways associated with cancer pathogenesis, as well as present alteration due to aging as demonstrated in Figure 2.

Along with the aging of blood cells, SIRT2 can control the oxidative stress derived from the expression of inflammasome factors, such as NLRP3 via controlling Caspase-1 levels that regulate cell death and the removal of dead cells [22]. This stress due to aging could also be controlled by SIRT3 that is linked to the increased production of antioxidant agents and mitochondrial proteins [23]. SIRT7 is involved in homeostasis derived from the interaction with NRF1 protein that consequently blocks mitochondrial damage and normalizes cellular respiration [24].

SIRT7 is localized in the q25.3 region of chromosome 17, which is commonly altered in several hematological diseases. In aging, SIRT7 levels are reduced when compared to young individuals [25]. The HSCs present senescence phenotypes during aging because of the reduction on SIRT7 expression. SIRT7 is attached to the initiation of the Transposon LINE1 pathway activation and, consequently, to cGAS-Sting, which is characterized by an increase in NF-kB, P65, TBK-1 and IRF3, as well as the loss of stability in heterochromatin [26].

Some studies have also evaluated the relationship between SIRTs and the effects on cellular repair. Tasselli et al. [27] identified that SIRT6 can deacetylate the H3K9ac, H3K56ac and H3K18ac, altering chromatin in the pericentric region in mice. Meanwhile, SIRT6 deficiency generates serious metabolic defects, ranging from genomic instability to acceleration of tumorigenesis. Minten et al. [28] reported the ability of SIRT2 to generate heterodimerization of BRCA1 with BARD1 and, consequently, the maintenance of the DNA repair pathway, allowing homologous recombination and consequent repair of double-stranded DNA lesions.

The role of SIRTs expression in the pathogenesis and overall survival (OS) of patients diagnosed with solid tumors is well known. The SIRT4 protein was noticeably increased in the breast cancer cells compared with adjacent non-tumor cells and, via multivariate analysis, SIRT4 represents an independent predictive factor of good prognosis for breast cancer patients [29]. SIRT3 induces apoptosis and necroptosis by regulating mutant p53 expression in small-cell lung cancer [30]. Low *SIRT6* expression was found to be associated with a better OS in breast cancer; on the other hand, it was found to be associated with a worse OS in gastrointestinal cancer [31]. In addition, the high expression of SIRT1 and SIRT2 are associated with poor prognosis in non-small cell lung cancer patients [32].

However, studies that seek to explain the role of these pathways in the hematopoietic aging process and the consequences of their instability in the pathogenesis of different onco-hematological diseases are still scarce. Lymphomas, myelomas, and leukemias are hematological malignances and an important cancer group with estimated incidences of ~30–80,000 and mortality of ~23% in 2022 in the United States [33]. In the most recent epidemiological report on the projection of new hematological cancer cases and deaths in the United States, it was estimated that, for the year 2022, approximately 184,000 new cancer cases were diagnosed, with an estimated ~58,000 cases of deaths for both sexes [33]. Therefore, this systematic review discusses the role of SIRTs in the advancement of pathogenesis and/or prognosis for different hematological cancer types. We also performed an evaluation SIRT gene expressions patterns on the Gene Expression Profiling Interactive Analysis (GEPIA) database in 30 cancer types to detect correlation in onco-hematological diseases.

## 2. Materials and Methods

### 2.1. Search Strategy

This systematic review was submitted and registered in the International Prospective Register of Systematic Reviews (PROSPERO) database (CRD42022310079). We conducted the present systematic review study based on the PICO strategy (patient or population, investigation/interest, and context/outcome) [34]. According to the concept proposed by the PICO method, the following aspects were included in this systematic review: cancer onco-hematologic (Population), expression of Sirtuins (Investigation/Interest) and pathophysiology (Context/Outcome). According to the three pillars defined for the establishment of the PICO, it was possible to define the main question of this review: What is the clinical relevance of Sirtuins gene expression in patients with onco-hematologic diseases?

### 2.2. Search Database

We developed the current systematic review based on the Preferred Reporting Items for Systematic Reviews and Meta-Analyses (PRISMA 2020) to establish the minimum evidence to carry out the systematic review about the role of SIRTs in the advancement of pathogenesis and/or prognosis for different hematological cancer types [35] (Figure 1). The authors systematically searched in the PubMed database to obtain high impact peer-reviewed studies (impact factor greater than or equal 2) published between 2011 and 2021 in the English language based on the following Medical Subject Headings (MeSH): “*sirtuins*” AND “*hematologic disease*” AND “*leukemia*” OR “*multiple myeloma*” OR “*lymphoma*”. The “*human*” filter was used to search for articles. Only research describing original articles was included in the study. Review studies, meta-analyses, comments, perspectives, editorials, or other research that did not provide original and unpublished results were excluded. All article records were screened by title and abstract by two independent authors (JVCG and LGC).

### 2.3. Gene Expression Profile Using GEPIA Database

GEPIA is a web-based tool (http://gepia.cancer-pku.cn/) of RNA sequencing data based on TCGA Research Network (The Cancer Genome Atlas Program) (https://www.cancer.gov/tcga) and GTEx (The Genotype-Tissue Expression) databases (https://gtexportal.org/home). The gene expression profile from the GEPIA [36] database was selected and tumor/normal differential expression levels for each SIRT was conducted via GEPIA tool.

## 3. Results

### 3.1. Search Results

Based on the MeSH terms used initially, this systematic review retrieved 393 article records (Figure 3). In the identification step by PRISMA, [35] a total of 219 articles were excluded for being duplicates (*n* = 69) and/or for other reasons, such as other diseases (*n* = 150). Thus, 174 articles were registered for the screening phase. According to the title and/or abstract of the remaining 117 articles, 103 articles not related to pathogenesis of hematological diseases were excluded. The complete files of the articles were obtained (*n* = 71). Of those 71 remaining articles, 41 studies related to research meta-analysis or review studies were excluded. Finally, a total of 30 articles were included in this systematic review study and the main findings are included in Table 1.

### 3.2. Lymphoid Neoplasms

#### 3.2.1. Acute Lymphocytic Leukemia (ALL)

Acute lymphocytic leukemia (ALL) is the most prevalent leukemia in children, accounting for at least 30 cases per million people in the United States in individuals under 20 years of age [37]. Several genetic factors are associated with the pathogenesis of this disease. In B cell ALL, chromosomal alterations such as t(12;21), t(1;19), t(9;21) and t(9;22) can be found [38,39]. The incidence of this disease in patients over 60 years of age is uncommon. The disease presents with greater risk factors in adult patients with the presence of the Philadelphia chromosome (Ph+). In this context, the incidence of the Philadelphia chromosome is higher in adult patients and its prevalence increases with age [40,41].

**Table 1 cancers-14-04611-t001:** Description of the SIRTs expression in oncohematological diseases.

Target	Disease	Main Findings	References
*SIRT7*	AML	Low expression in patients’ bone marrow cells	[25]
*SIRT7*	CML	Low expression in patients’ bone marrow cells	
*SIRT1*	CLL	High expression in the peripheral blood	[42]
*SIRT1*	MM	High expression in MM cell line	[43]
*SIRT1*	CML	High expression in primary human CML cells	[44]
*SIRT1*	CML	High expression in patient CML cells	[45]
*SIRT1*	CML	High expression in patient CML cells	[46]
*SIRT1*	Lymphomas	High expression in patients with Hodgkin’s lymphoma	[47]
*SIRT1*	AML	High expression of CD34+ and CD38- cells in the bone marrow	[48]
*SIRT1*	Lymphomas	High expression in cutaneous t-cell lymphomas	[49]
*SIRT1*	CML	High expression in patient CML cells	[50]
*SIRT1*	ALL	High expression in patient ALL cells and in cultured ALL lineage cells	[51]
*SIRT2*		High expression in patient ALL cells and in cultured ALL lineage cells	
*SIRT1*	CML	High expression in patient CML cells	[52]
*SIRT1*	Lymphomas	High expression in patients with Hodgkin’s lymphoma	[53]
*SIRT1*	Lymphomas	High expression in follicular hyperplasia	[54]
*SIRT1*	Lymphomas	High expression in cultured lymphoma lineage cells	[55]
*SIRT1*	MDS	Low expression in patient MDS cells	[56]
*SIRT1*	CML	High expression in patient CML cells	[57]
*SIRT1*	MM	High expression in MM cell line	[58]
*SIRT1*	T-ALL	High expression on T-ALL cells in vivo and in vitro	[59]
*SIRT1*	CLL	High expression in the peripheral blood	[60]
*SIRT2*		High expression in the peripheral blood	
*SIRT3*		High expression in the peripheral blood	
*SIRT4*		Low expression in the peripheral blood	
*SIRT5*		Low expression in the peripheral blood	
*SIRT6*		High expression in the peripheral blood	
*SIRT7*		High expression in the peripheral blood	
*SIRT1*	CLL	High expression in cultured CLL lineage cells	[61]
*SIRT2*	CLL	High expression in cultured CLL lineage cells	
*SIRT2*	AML	High expression in patient AML cells	[62]
*SIRT2*	MM	Low expression in patient MM cells	[63]
*SIRT3*	MM	Low expression in patient MM cells	
*SIRT2*	ALL	High expression on T-ALL cells both in vitro and in grafts	[64]
*SIRT2*	AML	High expression in patient AML cells	[65]
*SIRT3*	CLL	Low expression in patient CLL cells	[66]
*SIRT3*	Lymphomas	Low expression in mantle cell lymphomas	
*SIRT3*	AML	Low expression in patient AML cells	[67]
*SIRT6*	MM	High expression in patient MM cells	[68]
*SIRT6*	AML	High expression of CD34+ cells in the bone marrow	[69]
*SIRT6*	Lymphomas	High expression in diffuse B-cell lymphomas	[70]

Legend: ALL.: Acute lymphocytic leukemia. CLL.: Chronic lymphocytic leukemia. AML.: Acute myeloid leukemia. CML.: Chronic myeloid leukemia. MM.: Multiple myeloma. MDS.: Myelodysplastic syndromes.

The role of SIRTs in ALL is infrequently studied. Jin et al. described the role of Tenovin-6-mediated inhibition of SIRT1/2, an inhibitor of histone deacetylation activity, in inducing apoptosis in ALL. From the analysis of cells from patients with ALL and from cell lines of the REH and NALM-6 type, it was identified that the expression levels of SIRT1 and SIRT2 were increased in both samples. Furthermore, it was observed that Tenovin-6 inhibits the activity of these SIRTs, thus decreasing Wnt/β-catenin signaling and generating p53 hyperacetylation in cells (CD133+ and CD19-), in addition to being able to sensitize them for chemotherapy, such as etoposide and cytarabine [51].

Increased SIRT1 protein expression was correlated with increased NOTCH1 activation in vitro and in vivo [59]. The decreased expression of SIRT1 was capable of decreasing proliferation and colony formation of T-ALL cells; meanwhile, the high expression of the gene showed an opposite characteristic. In addition, the silencing of SIRT1 generated an increase in the lifespan of mice with T-ALL cells [59]. T-ALL leukemogenesis appears to be dependent on LMO2 activation triggered by SIRT2/NAMPT deactivation, a mechanism that generates increased stem cell hematopoiesis in the disease. In contrast, the inhibition of SIRT2 and NAMPT suppressed the growth of T-ALL cells both in vitro and in grafts [64].

#### 3.2.2. Chronic Lymphocytic Leukemia (CLL)

Chronic lymphocytic leukemia (CLL) is the most common leukemia in adults, accounting for 30% of all hematological diseases, with an incidence rate of 4.6 cases per 100,000 inhabitants in the United States [71]. This disease is characterized by a malignant transformation of B-lymphocytes and the consequent monoclonal accumulation of these cells in both peripheral blood and bone marrow, having a prognostic stratification where intermediate cases have up to eight years of survival and high-risk cases up to two years [72]. CLL is also characterized by distinct immunophenotypes, presenting deficiencies such as autoimmune hemolytic anemia (AIHA), immune thrombocytopenia (ITP), and immune neutropenia, in addition to the presence of CD5+, CD19+, CD20DIM, CD23+ cells [73].

In peripheral blood samples, *SIRT1*, *SIRT2*, *SIRT3*, *SIRT6* and *SIRT7* were found highly expressed in CLL patients when compared with samples from healthy patients, while *SIRT4* and *SIRT5* were negatively regulated [60,61]. However, it is controversial whether SIRT3 acts as an oncogene or a tumor suppressor in CLL. For example, a protein expression study has demonstrated that CLL patients’ cells show a high SIRT1 expression when compared to B lymphocytes from peripheral blood mononuclear cells [42]. On the other hand, Yu et al. [66] demonstrated that the loss of SIRT3 provides a growth advantage for B cell malignancies. Decreased SIRT3 was found to decrease IDH2 and SOD2 hyperacetylation by increasing the level of reactive oxygen species, suggesting that SIRT3 acts as a tumor suppressor in B cell malignancies. More studies are needed to establish the definitive role of SIRT3 in the pathobiology of CLL.

#### 3.2.3. Lymphomas

Lymphomas are a group of diseases that originated from B, T and NK cells that may present at different stages of maturation, with mature T-cell neoplasms accounting for at least 10% of all cases of cell lymphomas in this lineage. Lymphomas are classified as Hodgkin’s and Non-Hodgkin lymphomas, the latter being more common in elderly adults approximately 65 years of age [74]. Among non-Hodgkin lymphomas, diffuse large B-cell lymphomas (DLBCL) represent about 40% of all lymphoma cases. Hodgkin’s lymphomas are unusual lymphomas that affect young individuals; however, their prognosis is commonly favorable [75].

SIRTs are little explored in this type of pathology. Yu et al., demonstrated that *SIRT3* is decreased in mantle lymphoma samples, being correlated with a worse overall survival time. In addition, the decrease in *SIRT3* correlates with hyperacetylation of IDH2 and SOD2, decreasing the activity of these enzymes and consequently increasing ROS levels. This study demonstrated that this SIRT is suggested as an excellent biomarker and its overexpression presents as a tumor suppressor in B-cell malignancies [66]. *SIRT1* showed high nuclear expression in patient samples of Hodgkin’s lymphoma; once evaluated together with FOXP3 expression, SIRT1 inhibition can decrease the actions of T-reg cells [53]. In L-428 cells derived from Hodgkin’s lymphoma, the use of resveratrol decreased SIRT1, causing an increase in p53 and FoxO3a acetylation and an increase in apoptosis [47].

SIRT levels have also been explored in follicular lymphomas (FL) compared to DLBCL and follicular hyperplasia, indicating that in FL, *SIRT1* was more expressed compared to follicular hyperplasia and DLBCL. Cutaneous T-cell lymphomas showed the increased expression of SIRT1 [54]. The knockdown of *SIRT1* increases the FoxO3 and PARP cleavage in Jurkat cells by reducing the metabolic levels of these cells by up to 48 h, in addition to increasing apoptosis levels by increasing p53 activity [49].

*SIRT6* was shown to be highly expressed in DLBCL tumors and their high expression was directly linked to resistance to chemotherapy. In addition, their blockade was associated with sensitivity to chemotherapy and the increased percentage of apoptosis of these cells [70]. In rare B-cell lymphomas, such as primary effusion lymphoma, *SIRT1* has been shown to be necessary for the proliferation and maintenance of tumor cells, while SIRT1 inhibitors, such as Tenovim-6, efficiently decrease the neoplastic growth process [55].

#### 3.2.4. Multiple Myeloma

Multiple myeloma (MM) is the second most common monoclonal proliferation hematological malignancy in high-income countries. It accounts for approximately 1% of all cancers and at least 13% of all hematological cancers in the world. Its incidence is equivalent to 4.5–5 patients per 100,000 inhabitants and, in adults, the average age reaches 70 years with survival ranges from five to 10 years [76].

*SIRT1* is highly expressed in MM cell lineages when compared to other SIRTs, and its levels are associated with bortezomib resistance. Abnormal activation of the Hedgehog pathway has been shown to be critical for MM lineage cells [58].

*SIRT1* inhibitors proved to be efficient in the control of chemoresistance in patients with MM. SRT-1720 proved to be efficient as a specific inhibitor for SIRT1 and was able to prevent the growth of MM cells by generating cytotoxicity in cells of the same lineage. In addition, it was found that the SIRT1 inhibitor was able to activate the caspases 8 and 9, triggering the cleavage of caspase 3, increasing the levels of apoptosis of these cells [43]

The expression of *SIRT2* and *SIRT3* is reduced in MM cells when compared to healthy controls. Its low expression is correlated with more advanced clinical stages of the disease and redox imbalance, suggesting that both genes are excellent biomarkers in disease stratification [63].

*SIRT6* is positively regulated in MM cells, but its levels are virtually undetectable in peripheral blood. In murine models, SIRT6 knockout leads to increased DNA damage and genomic instability. In MM cell lines, SIRT6 is able to oppose cell growth in this cell line, opposing the pathway of the protein kinase activated by myogen (MAPK). In the face of double-chain damage, SIRT6 blocks MAPK, resulting in the increased phosphorylation of Chk1 in Ser317, maintaining the genome and correcting DNA damage [68].

### 3.3. Myeloid Neoplasms

#### 3.3.1. Acute Myeloid Leukemia (AML)

Acute myeloid leukemia (AML) is a neoplasm that presents immature myeloid cells. Here, age and the presence of myelodysplastic syndromes (MDS) are among the main risk factors, having a survival average of five years in at least 25% of cases [77]. Clonal expansion of HSCs is due to genetic changes that fall into three categories: chromosomal aberrations, multiple gene mutations, and epigenetic changes [78].

In 2016, the World Health Organization’s (WHO) classification system was updated. Currently, AML presents several diagnostic parameters, including recurrent genetic abnormalities, such as translocations of chromosomes 15 and 17 t(15;17), t(8;21), t(6;9), in addition to inversions of chromosomes 3 and 16 [79,80].

Increased protein expression of SIRT1 was identified in CD34+ and CD38- cells in bone marrow samples from patients with AML. When compared with normal samples, *SIRT1* expression was increased in high-risk and intermediate-risk patients when compared to low-risk patients [48].

An increase in SIRT6 expression was identified in CD34+ cells of patients with AML. On the other hand, the decrease of SIRT6 expression increases sensitivity to DNA-harmful agents such as daunorubicin and cytarabine, in vitro and in vivo [69].

The chromosomal position of SIRT7 (17q25.3) is commonly altered in acute leukemias. In the bone marrow of patients with AML, the level of SIRT7 expression is reduced when compared to healthy individuals. Studies indicate that *SIRT7* is an excellent biomarker in response to stem cell treatment with pharmacological inhibitors of driver mutation. Its low expression is linked to age-related disorders, especially in patients with a mutation in *FLT3*; in this case, treatments in cells with *FLT3-ITD* mutations showed an increase in SIRT7 expression and the consequent improvement of patients [25].

SIRT3 decreases ROS production by deacetylating SOD2 and IDH2 and can be a crucial marker for the leukemogenesis process. In AML, SIRT3 expression is relatively lower and varies according to the subclassification of the disease. For that, it can be difficult to predict its exact expression; however, its activity is more focused on the regulation of ROS than necessarily in its expression [67].

SIRT2 is positively regulated in AML. This gene is presented as unfavorable in the prognosis of AML, as its expression is higher in relapses when compared to newly diagnosed patients [65]. SIRT2 is closely related to NAMPT, an enzyme that increases SIRT2 activity by increasing NAD+ synthesis. Dan et al. demonstrated that the inhibition of SIRT2 or NAMPT increased apoptosis of AML blasts and reduced the proliferation of these cells [62].

#### 3.3.2. Chronic Myeloid Leukemia (CML)

Chronic myeloid leukemia (CML) is probably the most studied malignant disease in humans, accounting for at least 20% of all diagnoses of leukemias in adults [81]. It has a total of total of specific phases, ranging from a chronic phase, through what we call the accelerated phase, reaches a phase of blast crisis, and can end with metastasis, organ failure and death [82].

Its main genetic abnormality is focused on the translocation of the ABL1 gene on chromosome 9 at the q34 position with the BCR gene region on chromosome 22 at the q11 position, resulting in the abnormal formation of a protein (BCR-ABL) whose effect is linked with abnormal cell proliferation in the bone marrow [83].

Studies relating SIRT genes to CML are scarce. Currently, only *SIRT1* is well analyzed in this disease. Being highly expressed, this gene is closely related to the maintenance of CML leukemic stem cells (CML-LSC) [46]. The reduction of its expression by the RNAi leads to the increased induction of apoptosis, the stopping of the cell cycle and increased sensitivity to damage by etoposides, besides reducing tumorigenesis in the CML cell line [52]. SIRT1 is activated with the BCR-ABL transformation and this increase in transcription is linked to the signal transducer and transcription activator 5 (STAT5). In contrast, the knockout of SIRT1 led to the decreased growth of CML cells and similar diseases in the rat model. [44].

The effect of shRNA in *SIRT1* in CML is reducing CML-LSC levels in mice, while increasing sensitivity to tyrosine kinase inhibitors. In addition, reducing SIRT1 levels reduces mitochondrial respiration in these cells [45,46]. Studies with Hsp90 protein inhibitors and SIRT1 inhibitors have identified that both have a characteristic therapeutic effect, allowing greater sensitivity to recurrent therapies for CML cells that have resistance to multiple drugs [50]. High levels of SIRT1 can reduce the level of inflammation caused by lipopolysaccharides in CML k562 cells, reducing toll-like 4 receptor (TLR4) sensors relevant to inflammation, as well as the reduction of NF-kB and p65, causing a decrease in ROS [57]. Kaiser et al. described the expression of *SIRT7* in CML samples. In this group, SIRT7 presents a decreased expression when compared to healthy individuals; however, this expression varies with age, demonstrating a marked decrease in expression in younger individuals (20–39 years) [25].

#### 3.3.3. Myelodysplastic Syndrome (MDS)

Myelodysplastic Syndrome (MDS) is a set of clonal diseases of hematopoietic stem cells present in the bone marrow, being characterized by hematopoietic deficiency, causing dysplastic morphologies of hematopoietic elements and peripheral cytopenias, with a high probability of progression to AML [84,85].

Recent studies have identified several markers of mutations in MDS, including splicing factors, epigenetic regulators, cohesins, transcription factors, and translation signals [80,86,87]. In an analysis of 3324 patients diagnosed with MDS in Latin America with a mutation in the TP53 gene, at least one third presented monoallelic mutation characteristic for the gene, this being characterized as a high-risk factor for the disease [88].

The main risk factor for the disease is age. Belli and other collaborators demonstrated that the average age of patients diagnosed with this disease in Latin American countries is around 69 years, where 75% of patients are over 60 years of age at the time of diagnosis and the incidence practically doubles every 10 years in patients over 40 years of age [89].

Studies using SIRTs as biomarkers for MDS are uncommon. Recently, a study published in 2018 broke the paradigm that existed between SIRT and MDS. When analyzing the expression of *SIRT1* in MDS patients, the study demonstrated that, in view of the increased presence of two microRNAs (*miR-34a*), a SIRT1 mRNA blockade occurs, increasing the acetylation levels of the TET2 protein (Figure 4). This increase in acetylation decreases DNA methylation control and consequently contributes to the growth and maintenance of MDS-HSPCs [56].

Yamakuchi et al. [90] evaluated the relationship between *miR-34a* and *SIRT1* in vitro. From events such as telomere shortening, oxidative stress and DNA damage, a positive feedback loop between p53 activation and the overexpression expression of *miR-34a* caused a decrease in SIRT1 and consequent increase in p53 acetylation. Therefore, that leads to leading to increased apoptosis and cell cycle arrest. Both studies demonstrate the relationship between the increase in *miR-34a* expression and the decrease in *SIRT1*, suggesting that p53 activation is one of the factors for such an effect in MDS (Figure 4).

### 3.4. Sirtuin Gene Expression Profile in Hematological Malignances

We conducted a detailed analysis on cancer versus healthy status of the SIRTs gene expression in 30 cancer types (GEPIA database). Among all types of cancers detailed in our analysis by GEPIA, we describe only the findings of SIRTs expression in hematological malignancies, more specifically in AML and Lymphoid Neoplasm Diffuse Large B-cell Lymphoma (DLBCL) (Appendix A).

We observed that mRNA expression of *SIRT1*, *SIRT3*, *SIRT5* and *SIRT6* genes were upregulated in DLBCL (*n* = 47) when compared to normal samples (*n* = 377). We did not identify differential expression of SIRTs in relation to the overall survival with DLBCL. On the other hand, high *SIRT6* expression was associated with a poor AML overall survival (log-rank *p* = 0.0066) (Appendix A).

## 4. Discussion

This systematic review analyzed 30 studies that aimed to evaluate the role of SIRTs in the pathobiology of different malignant hematological diseases. Although a vast group of publications allow us to understand the role of SIRTs in oncohematological diseases, we identified that the understanding of the importance of these genes is still lacking in a complex group of oncological diseases, such as those of blood tissue. Leukemias were the group of blood malignance disorders with the greatest molecular description of the importance of SIRTs in the advent of these diseases. CLL was one of the few diseases that showed the general expression of SIRTs in patient samples, with SIRT1 being the most studied target, both in pathophysiological factors and in therapeutic approaches [59].

AML showed more results with important characterization of gene expression and key knowledge of the therapeutic targets [46,69]. In addition, AML has more results in the GEPIA database, easing the in silico evaluation of SIRTs. SIRT7 has been shown to be an excellent biomarker of response to treatments with inhibitors of driver mutations in this disease. In addition, SIRT3 has shown a close correlation with leukemogenesis, even though its relationship with the disease is due to its activity rather than its expression [25,67]. *SIRT1* is the most studied gene in the majority of the diseases addressed in this study. In AML, this gene is considered an excellent prognostic marker and is highly expressed in patients at high risk and intermediate risk [48]. In addition, its role in the clearance of neoplastic cells through caspases and by induction of cellular cytotoxicity is already elucidated, demonstrating the importance of this gene in therapeutic contexts [58].

The general results of leukemia have shown that: (i) *SIRT2* and *SIRT3* can be considered as important prognostic markers for the disease, (ii) in ALL, SIRT2 participates in processes such as tumor growth and (iii) in CLL, SIRT3 acts in the control of reactive oxygen species [51,66]. *SIRT2* plays an important role in several biological processes, such as cell cycle control, genomic integrity, microtube dynamics, cell differentiation, metabolic networks, and autophagy [16]. *SIRT2* was able to control inflammasome pathways and also inflammatory levels in hematopoietic stem cells, in addition to controlling oxidative stress via NLRP3-Caspase1, acting in the control of cell clearance [22]. Therefore, *SIRT2* occupies an important point of discussion in this study group both at the level of pathogenesis and in relation to therapeutic possibilities.

Although there is no complete knowledge of the molecular role of SIRTs, there is already relevant data capable of introducing efficient research in clinical and therapeutic contexts for diseases, such as lymphomas and multiple myeloma [43,49]. Characteristics such as neoplastic cell maintenance and tumor growth have been well documented in these cases, especially in studies with SIRT6 in correlation with pathways such as AMPK [68]. *SIRT6* acts in regulating DNA repair, telomere protection and genome stability [16]. *SIRT6* has shown activity in cellular chemoresistance, however its blockade is related to increased sensitivity to chemotherapy and cellular apoptosis in diffuse large B-cell lymphomas [70].

In lymphomas, *SIRT1* levels also showed correlation with the effects of cell maintenance, mainly by inhibition of the p53 pathway. High protein levels of SIRT1 lead to increased protein deacetylation of p53 and the consequent blockade of its activity; however, the decreased protein level of SIRT1 (due to the influence of HDACs inhibitors) causes the increased acetylation and consequent hyperactivity of p53, leading to increased levels of cellular apoptosis [49,90].

MDS was the group of diseases where the role of SIRTs is less understood. This specific field of oncohematology still requires precise studies that align the SIRTs activity pathways in the pathophysiological mechanism of the disease. The expression analysis of *SIRT1* combined with the acetylation profile of TET2 paves the way for an idea whose precursor factor for the disease is by non-mutational mechanisms of TET2, in addition to the pathognomonic pathways already established for the disease [56]. In an in silico approach, we conducted an analysis in the GEPIA database, and we validated the results of the present systematic review when we observed that the *SIRT1*, *SIRT3*, *SIRT5* and *SIRT6* genes are also up-regulated in onco-hematological diseases such as DBLC when compared with normal tissues.

It is important to highlight that some studies aimed to understand a method to control the activity of SIRTs for the treatment of cancer and the control of aging [91,92]. The SIRTs have crucial roles in the understanding of pathologies such as obesity, neurodegenerative diseases, cardiovascular diseases, inflammation and cancer [93]. Because of this, modulating the activity of SIRTs presents itself as an extremely important factor for the administration of new methodologies for treatment and disease control.

Among liquid tumors, few studies have demonstrated the relationship between SIRTs activators and oncohematologic diseases. RUI et. al. [94] showed that by inducing resveratrol (RSV), a potent activator of SIRT1, in irradiated mice that presented medullary deficiency (pancytopenia). It was possible to induce an increase in cell populations and led to the consequent reversal of medullary failure. In human MM cell lines, RSV activated IRE1α by the XBP1 messenger RNA splicing and the phosphorylation of IRE1α in MM cells. Associated with its cytotoxicity, RSV decreased the DNA-binding capacity of XBP1 and increased the enrichment of SIRT1 in the binding region in the XBP1 promoter; moreover, it selectively suppressed the transcriptional activity of XBP1s (MM developmental factor), while stimulating gene expression of the IRE1/XBP1 pathways (the pathway responsible for endoplasmic reticulum stress homeostasis) [95].

Other activators have also been shown to be effective in the modulating context of SIRTs. The SIRT1 activator SRT1720 and the SIRT1 agonist SRT2104 were able to inhibit HSPCS MDS colony formation in murine models by SIRT1-induced TET2 acetylation reversing the dysplastic phenotype, moreover, the expression of *SIRT1* and TET2 target genes was also positively regulated in MDS CD34+ cells after SRT1720 treatment [56]. Patients with Fanconi Anemia, a disease characterized by bone marrow failure, MDS and AML, have decreased HSPCs and progressive loss of these cells. Also, compounds that can activate SIRT1 (SRT3025) were able to correct the HSPCs deficiency, demonstrating the relationship of *SIRT1* with normal hematopoiesis [96].

Yuan et al. [44] demonstrated the inhibitory effects of Sirtinol and Tenovin-6 in CML cells with increased SIRT1 expression. These inhibitors block SIRT1 activity leading to increased sensitivity of CML cells to imantinib [97]. The inhibitory activity of SIRT1 by Tenovin-6 has also been described in CML studies on p53 activation, as well as on the c-MYC oncogene pathway. Furthermore, human AML FLT3-ITD stem cells showed sensitivity to Tenovin-6. The sum of these studies demonstrates the inhibitory efficiency and the broad field of study of modulating SIRTs in oncohematological diseases [45,98].

## 5. Conclusions

Our systematic review demonstrated that *SIRTs* are primarily upregulated in oncohematological diseases such as AML, ALL, CLL, MM and lymphomas. Only *SIRT3* and *SIRT7* were downregulated in lymphoma, AML and CML. SIRTs differential gene expression was also validated in in silico analyses in DLBC and AML. These results reinforce the importance and clinical relevance of SIRTs in the pathogenesis and follow-up of onco-hematologic patients.

Sirtuins are presented as a broad group of genes that act in various metabolic and cell maintenance activities; however, their activities may be altered in the most diverse oncohematological diseases. We demonstrated that recent scientific evidence supports the role of SIRTs in the pathobiology of hematological malignances, such as leukemias, lymphomas and myeloma. The need for further high-quality research and clinical trials that expands the knowledge around SIRT inhibition and its effect on controlling clonal progression caused by the genomic instability characteristics of these diseases is clear.

Thus, we conclude that these genes may be considered as important markers of pathobiology and in the treatment in oncohematological diseases; however, it is necessary to investigate the role of these genes in unknown pathways, especially in diseases whose information is still scarce, such as an MDS. Finally, SIRTs represent potential molecular targets in the control of the effects caused by aging on the failures of the hematopoietic system that can lead to the involvement of hematological neoplasms.

## Figures and Tables

**Figure 1 cancers-14-04611-f001:**
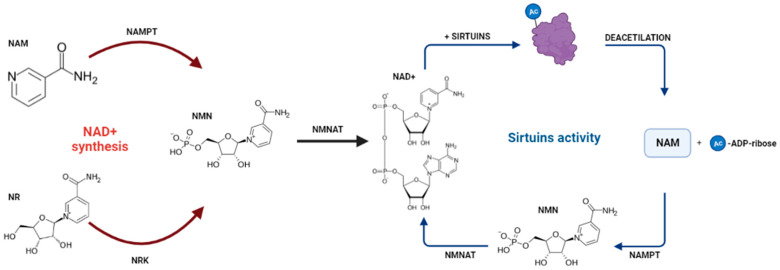
In the process of NAD+ formation, Free Nicotinamide (NAM) is converted to Nicotinamide (NMN) by the action of the enzyme Nicotinamide phosphoribosyltransferase (NAMPT). The same result is obtained when supplemental Nicotinamide Riboside (NR) is converted to NMN by the enzyme Nicotinamide Riboside Kinase (NRK). NMN is converted to NAD+ by the enzyme NMN Adenyltransferase (NMNAT). Upon binding with the SIRTs, NAD+ is hydrolyzed, the acetyl group attached to the protein is removed, deacetylating the protein and, as a result, NAM and 2’−O−acetyl−ADP−ribose are released. Finally, NAM returns to the NAD+ formation cycle, restarting the cycle.

**Figure 2 cancers-14-04611-f002:**
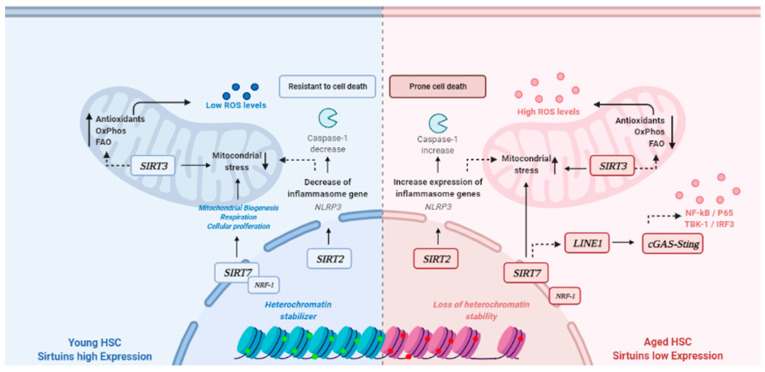
Sirtuins activity pathways in young versus aged HSCs. Legend: FAO: Fatty acid β-oxidation. HSC: Hematological Stem Cells. OXPHOS: Mitochondrial proteins oxidative phosphorylation system.

**Figure 3 cancers-14-04611-f003:**
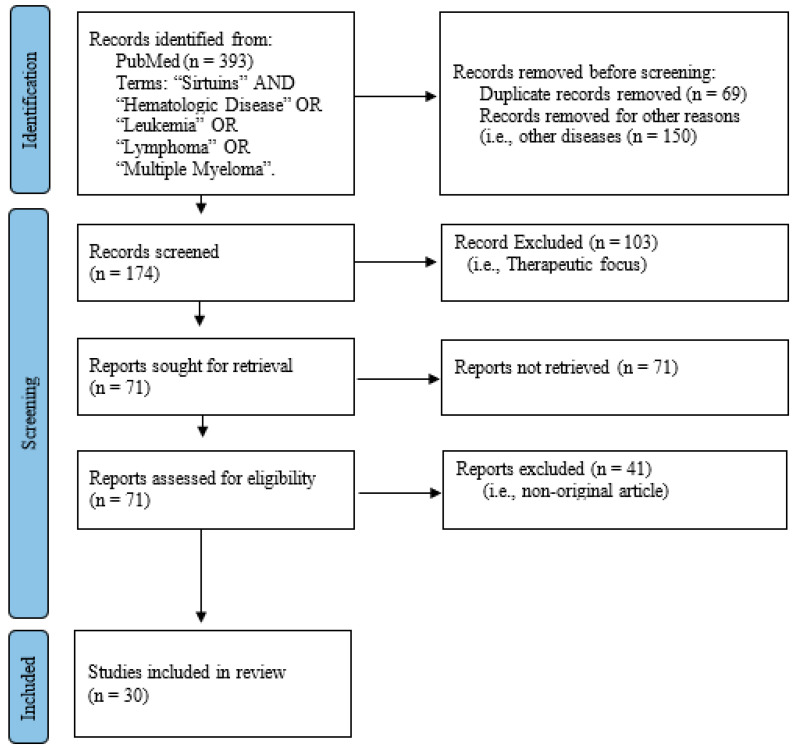
Flowchart of data obtained from the search of PUBMED/Medline records based on the PRISMA methodology.

**Figure 4 cancers-14-04611-f004:**
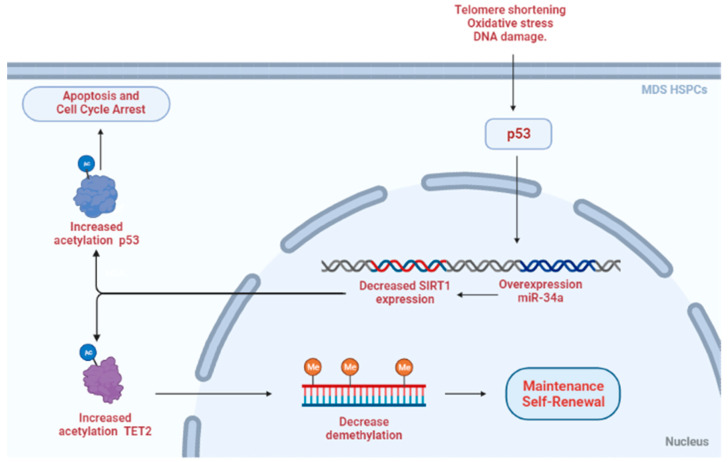
NAD+ synthesis and protein deacetylase activity by sirtuins in MDS.

## Data Availability

Not applicable.

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
