# Peer review of "Role of Sirtuins in the Pathobiology of Onco-Hematological Diseases: A PROSPERO-Registered Study and In Silico Analysis"

_cancers, 2022, doi:10.3390/cancers14194611_

Round 1

Reviewer 1 Report

The present manuscript reviews the information available of sirtuins in the pathogenesis of hematological neoplasms. This is an interesting review and useful information is shown. I have some comments aiming to improve the manuscript.

-          Results section should be organized considering myeloid and lymphoid diseases instead of acute and chronic diseases. It is difficult for a reader involved in diagnosis and treatment of hematological neoplasms to follow the present scheme.

-          In Results, paragraphs 2 and 3 should be reviewed since the role of SIRT3 in CLL seems to be contradictory.

-          In page 8, line 273 CML should be CLL; same page, line 278, CLL-B should be CLL.

-          In page 9, line 354, diffuse B-cell lymphomas should be diffuse large B-cell lymphomas. The correct acronym is DLBCL, instead of DLBC.

-          In page 10, what do the authors mean when they say?: “Multiple myeloma (MM) is one of the most monoclonal proliferation hematological diseases in the world.”

-          In Discussion, the authors should elaborate mores the first 2 sentences, as well as the second paragraph, that contains grammar mistakes.

Reviewer 2 Report

The authors present an interesting review on the "Role of Sirtuins in the pathobiology of Onco-Hematological diseases: a PROSPERO-registered study and in silico analysis" The present article is of considerable interest and well-conducted,  however, I have some concerns which in my humble opinion should be addressed prior to publication:

1) the major drawback in the present manuscript lies in the fact that neither SIRT activation nor SIRT inhibition has been discussed in depth. There are some  SIRT activators or inhibitors depending on the SIRT isoform described in the literature. Some of these modulators have been tested also in liquid cancers, thus I suggest that a section should be added to the review dealing with this approach. (see also point 2) Partially has been said something in the conclusion section, however, only inhibition is mentioned. As rightly stated by the authors very likely SIRT3/SIRT5 activation is probably desirable in numerous cancers, specific activation by a small molecule would be highly interesting. 

2) Table 1 is summarizing the SIRT isoform expression in various liquid cancers. The table would benefit from an additional column stating whether SIRT activation or SIRT inhibition would be beneficial and in case if already tested cited the relevant literature

Minor point: there are some typos present in the manuscript

Round 2

Reviewer 1 Report

I do not have additional comments

Author Response

We appreciate the positive feedback from the reviewer.

Reviewer 2 Report

Even though I do not fully agree with the authors for the reason why the SIRT activators are practically omitted, the rest of the review is of considerable interest to be published. The remaining points of my concerns have been answered.

Author Response

(The authors gave the same response as above.)
